# Comparative Chloroplast Genome Study of *Zingiber* in China Sheds Light on Plastome Characterization and Phylogenetic Relationships

**DOI:** 10.3390/genes15111484

**Published:** 2024-11-19

**Authors:** Maoqin Xia, Dongzhu Jiang, Wuqin Xu, Xia Liu, Shanshan Zhu, Haitao Xing, Wenlin Zhang, Yong Zou, Hong-Lei Li

**Affiliations:** 1Chongqing Engineering Research Center for Horticultural Plant, College of Smart Agriculture, Chongqing University of Arts and Sciences, Chongqing 402160, China; xiamq@cqwu.edu.cn (M.X.); jiangdongzhu11@163.com (D.J.); liuxiavip8@163.com (X.L.); xinght@cqwu.edu.cn (H.X.); zhangwenlin88519@126.com (W.Z.); nevernever107@126.com (Y.Z.); 2Zhejiang Lab, Hangzhou 311500, China; xuwuqin@zhejianglab.com; 3State Key Laboratory for Managing Biotic and Chemical Threats to the Quality and Safety of Agro-Products, Ningbo University, Ningbo 315211, China; zhushanshan1@nbu.edu.cn; 4Chongqing Key Laboratory for Germplasm Innovation of Special Aromatic Spice Plants, College of Smart Agriculture, Chongqing University of Arts and Sciences, Chongqing 402160, China

**Keywords:** chloroplast genomes, *Zingiber*, phylogeny, comparative analyses

## Abstract

Background: *Zingiber* Mill., a morphologically diverse herbaceous perennial genus of Zingiberaceae, is distributed mainly in tropical to warm-temperate Asia. In China, species of *Zingiber* have crucial medicinal, edible, and horticultural values; however, their phylogenetic relationships remain unclear. Methods: To address this issue, the complete plastomes of the 29 *Zingiber* accessions were assembled and characterized. Comparative plastome analysis and phylogenetic analysis were conducted to develop genomic resources and elucidate the intraspecific phylogeny of *Zingiber*. Results: The newly reported plastomes ranged from 161,495 to 163,880 bp in length with highly conserved structure. Results of comparative analysis suggested that IR expansions/contractions and changes of repeats were the main reasons that influenced the genome size of the *Zingiber* plastome. A large number of SSRs and six highly variable regions (*rpl20*, *clpP*, *ycf1*, *petA*-*psbJ*, *rbcL*-*accD*, and *rpl32*-*trnL*) have been identified, which could serve as potential DNA markers for future population genetics or phylogeographic studies on this genus. The well-resolved plastome phylogeny suggested that *Zingiber* could be divided into three clades, corresponding to sect. *Pleuranthesis* (sect. *Zingiber* + sect. *Dymczewiczia*) and sect. *Cryptanthium*. Conclusions: Overall, this study provided a robust phylogeny of *Zingiber* plants in China, and the newly reported plastome data and plastome-derived markers will be of great significance for the accurate identification, protection, and agricultural management of *Zingiber* resources in the future.

## 1. Introduction

*Zingiber* Mill. (family Zingiberaceae, tribe Zingibereae) is an economically important herbaceous perennial genus. Species of this genus are mainly distributed in tropical to warm-temperate Asia [1,2]. In China, there are about 42 *Zingiber* species [1], many of which have been cultivated and utilized for thousands of years for their uses as herbal medicines [3,4], spices [5], and landscaping [6]. For example, the most famous one is *Zingiber officianle* (commonly known as ginger), which has been cultivated since the Spring and Autumn period (770–476 BC). It is widely used in traditional Chinese medicine due to its efficacy in treating colds, emesis, and coughs [7,8]. In addition, gingerols of ginger play an important role in inhibiting hyperplasia, inflammation, and carcinogenesis [9,10]. Shampoo ginger (*Zingiber zerumbet*) has viewable red inflorescence. Its rhizome can cure stomach pain, diarrhea, swelling, and inflammation, and its aromatic oil can be extracted as a raw material for flavoring [1,11,12]. Young inflorescences of *Zingiber mioga* and *Zingiber striolatum*, called ‘Yangho’, are popular healthy vegetables and are also used to cure cough, indigestion, and constipation [1,13]. In addition to the species described above, *Zingiber* has received increasing attention in China and around the world, showing great potential in the food and pharmaceutical industries.

Due to the highly morphological similarity of closely related *Zingiber* species, especially during non-flowering seasons, misuse and purposeful adulteration of congeneric plants keep coming out in the commercial products that utilize them as the raw material [7,14,15,16,17]. Additionally, excessive exploitation of wild resources and destruction of their habitats by local people have threatened the diversity of the genus. To address these problems, accurate identification of *Zingiber* species is very essential for the manufacture of effective and safe medicines, as well as the protection of wild germplasm resources [18,19].

The most recognized classification system for *Zingiber* is based on the features of the inflorescence, dividing the genus into four sections: (1) sect. *Zingiber*, characterized by a spike on a long erect peduncle (Figure 1a–c); (2) sect. *Cryptanthium*, with radical inflorescences composed of a spike appearing at ground level with a subterranean peduncle (Figure 1d–f); (3) sect. *Pleuranthesis*, featuring inflorescences on peduncles emerging laterally from the leaf sheaths (Figure 1g); (4) sect. *Dymczewiczia*, with terminal inflorescences (Figure 1h) [20,21,22]. However, an increasing number of studies have proposed that sect. *Dymczewiczia* should be amalgamated with sect. *Zingiber* due to variability in the position of the inflorescence in certain species [22], pollen morphology [23], and phylogenetic relationships based on several single-loci sequences [2,24]. Moreover, the relationships among sect. *Cryptanthium*, sect. *Pleuranthesis,* and sect. *Zingiber* have always been mysteries. A parsimonious tree based on *ITS* sequences from 23 *Zingiber* species suggested that sect. *Cryptanthium* was the first divergent group and formed a sister group to other sections [2]. However, the *ITS* phylogeny obtained by Bai et al. [24] and the chloroplast phylogeny from Jiang et al. [6] indicated that sect. *Pleuranthesis* was the basal group within *Zingiber*. Such uncertainty in phylogeny may be caused by extensive hybridization/introgression among *Zingiber* species [24]. Obviously, in order to solve these long-standing problems in the classification of *Zingiber*, it is essential to conduct a comprehensive phylogenetic study with efficient molecular markers and advanced analytical methods.

In recent years, the plastid genome (plastome), as a kind of super-barcode, has been increasingly applied to the classification of taxonomically complex plant groups (e.g., *Allium* [25]; *Ilex* [26]; *Polygonatum* [27]; *Cymbidium* [28]). Plastome is usually uniparentally inherited and exhibits a typical quadripartite structure with about 150 kb in genome size, which can bring a hundred-fold increase in genetic information sites compared to the standard DNA barcode and therefore enhance the taxonomic resolution power [29,30]. Comparative plastome analyses of many medicinal plant genera have found that plastomes contain a lot of genetic variation, such as microsatellites, single nucleotide polymorphisms (SNPs), repeat sequences, and divergent hotspot regions, which are potential molecular marker resources for population genetics or phylogeography research [31,32,33]. Owing to continuous technological improvement in next-generation sequencing (NGS), plastomes can be generated more cheaply and easily. Collectively, the plastome has been recognized as the next generation of plant DNA barcoding with an improved species discrimination rate and reliable phylogenetic resolution [33,34].

In this study, we report 29 newly sequenced plastomes from the genus *Zingiber*. The objectives of this study were to (1) describe the characteristics of plastome variations in *Zingiber* through comparative methods; (2) identify useful genetic resources like plastome-derived SSRs and divergent hotspots; and (3) build a well-resolved phylogenetic backbone for *Zingiber* species in China. The results of the present study are predicted to enhance our understanding of the phylogenetic relationships and help to accurate identification, conservation, and agricultural management of *Zingiber* resources.

## 2. Materials and Methods

### 2.1. Sampling, DNA Extraction and Genome Sequencing

Plant materials of 29 accessions representing 17 *Zingiber* species from China were collected in this study (Table 1). With the exception of *Z. atroporphyreum*, *Zingiber cochleariforme*, *Z. ellipticum*, *Zingiber gulinense*, *Z. purpureum*, *Zingiber officinale*, and *Z. striolatum*, which were sampled from liquid nitrogen frozen fresh leaves, the remaining samples were taken from leaves of herbarium specimens. Total genomic DNA (gDNA) was extracted from *c.* 50 mg of leaves using the Plant Genomic DNA Kit (TIANGEN, Beijing, China). After sample QC, gDNA was fragmented by ultrasound on Covaris E220 (Covaris, Brighton, UK). Fragments from 300 bp to 500 bp were end-repaired and A-tailed, then ligated indexed adaptors on both ends. The products were amplified by PCR and circularized to get a single-stranded circular (ssCir) library. Finally, the ssCir library was amplified through rolling circle amplification (RCA) to obtain DNA nanoball (DNB) and sequenced by the MGI-DNBSEQ platform (Shenzhen, China) to generate 150 bp paired-end reads.

In addition, we downloaded the published plastomes of Zingiberaceae and outgroups from GenBank for subsequent phylogenetic analyses. In total, 45 accessions presented 42 species were included, containing 19 *Zingiber* species, 17 additional Zingiberaceae species, two Costaceae species, one Cannaceae species, and three Musaceae species (Appendix A).

### 2.2. Assembly and Annotation of Plastome

All raw data were trimmed by removing low-quality reads and adapters in Trimmomatic v0.39 [35], then used for plastome de novo assembly in GetOrganelle v1.7.5 [36]. The *get_organelle_from_reads.py* can automatically estimate reads required for assembly. We used 15 rounds (-R 15) of extension iterations to obtain a complete plastome or to stabilize the incomplete plastome result. The *k*-mers (-k) were set as 21,45,65,85,105. The assembled plastome circular sequences were annotated using Geneious R9 (https://www.geneious.com/updates/geneious-prime-r9-1 (accessed on 20 August 2023)). All 29 plastomes were aligned using the MAFFT v.7 plugin with the previously published plastome of *Z. officinale* (MW602894) as the reference [37]. Then, reference annotations were transferred to these newly assembled plastomes and manually checked for the accuracy of exon/intron boundaries and start/stop codon positions. The structure map of *Zingiber* plastomes was visualized using the software Chloroplot [38], followed by manual editing.

### 2.3. Comparative Analyses of Plastome

Whole plastome comparation of 29 *Zingiber* accessions was performed through the online software mVISTA under a ShufeLAGAN mode by applying the annotation of *Z. atroporphyreum* 1 as the reference (http://genome.lbl.gov/vista/mvista/submit.shtml (accessed on 12 February 2024)) [39]. We used IRscope to detect expansions or contractions in the inverted repeat (IR) regions of 29 *Zingiber* plastomes (https://irscope.shinyapps.io/irapp/ (accessed on 15 February 2024)) [40]. The software can visualize the junctions between the IRs and the large single copy (LSC)/small single copy (SSC) regions.

### 2.4. Identification of Plastid Microsatellites and Repeats

Microsatellites of each sequenced plastome were detected using MISA-web [41]. Thresholds of the mono-, di-, tri-, tetra-, penta-, and hexa-nucleotide microsatellites were set as ten, five, four, three, three, and three repetitions, respectively. The number and position of plastid repeat sequences were identified by REPuter [42]. Both forward, palindromic, complement, and reverse types were included, while only repeats with a length larger than 30 bp, sequence identity over 90%, and a hamming distance of 3 were identified.

### 2.5. Nucleotide Diversity Analyses of Plastome

To identify highly divergent regions in the *Zingiber* plastome, we calculated nucleotide diversity (*π*) of genes (CDS, tRNAs, rRNAs) and intergenic spacers (IGS) that contained >1 mutation site and >100 bp aligned length in 29 complete plastomes of *Zingiber*. These plastome regions were aligned using MAFFT v.7, respectively, and then calculated *π* with DnaSP v5.10 [43]. Furthermore, the genetic distance of each selected hyper-variable region and their combinations was calculated using MEGA 6.0 with the K2P distance model to verify their potential as plastid DNA barcodes [44].

### 2.6. Phylogenetic Inferences

In order to provide a most complete plastid phylogenetic tree for *Zingiber*, we conducted phylogenetic inferences using 74 complete plastomes (Table 1 and Appendix A). A total of 31 *Zingiber* species (52 accessions) were included, which represent 73.81% of the total number of *Zingiber* species in China. All plastomes were aligned to concatenate into a supermatrix in MAFFT v7 [37]. Phylogenetic analyses were performed using both maximum likelihood (ML) and Bayesian inference (BI) methods. The ML analyses were carried out by IQ-TREE v1.6.12 [45] with the best substitution model and partitioning scheme simultaneously implemented in ModelFinder under the Bayesian information criterion (BIC) [46,47]. The MrBayes XSEDE v3.2.7 on the CIPRES Science Gateway was used for the BI analysis [48]. Two parallel runs of four Markov chain Monte Carlo (MCMC) chains were run for two million generations, with a sampling frequency of once every 1000 generations. The first 10% of resulted trees were discarded as burn-in, and the remaining trees were used to build the consensus tree and obtain associated clade posterior probabilities (PPs).

## 3. Results

### 3.1. Characteristics of Newly Assembled Plastome

In this study, 29 plastomes of *Zingiber* species were reported (GenBank accessions: OR337869–OR337880; CNGB accessions: N_001486761–N_001486771, N_001486773–N_001486778), with sequence lengths ranging from 161,495 bp (*Z. purpureum* 1) to 163,880 bp (*Z. atroporphyreum* 1 and 3) (Table 1; Figure 2). Like the plastome of most angiosperms, they displayed the typical quadripartite structure and were composed of a large single-copy (LSC; 87,486–88,460 bp) region, a small single-copy (SSC; 15,577–19,488 bp) region, and a pair of inverted repeats (IRs; 27,035–29,929 bp) (Figure 3). Plastome GC contents of them ranged from 35.8% to 36.2% (Table 1). All plastomes included 132 genes arranged in the same order, of which 112 were unique genes, containing 78 protein-coding gene sequences (CDS), 30 tRNA genes, and four rRNA genes. Of all the genes, nine CDS (*atpF*, *ndhA*, *ndhB*, *petB*, *petD*, *rpl2*, *rpl16*, *rpoC1*, and *rps16*) and six tRNA (*trnA*, *trnG*, *trnI*, *trnK*, *trnL*, and *trnV*) genes contained only one intron, whereas three genes (*clpP*, *rps12*, and *ycf3*) possessed two introns (Appendix A).

### 3.2. Plastome Variations Within Zingier

Whole plastome comparation of mVISTA has not discovered any structural variation for the reported plastomes (Figure 4). With *Z. atroporphyreum* 1 as the reference, the consistency of most plastome regions was over 90%, and the divergence within translated regions (exons) was much higher than that of conserved non-coding sequences (NCSs) and untranslated regions (UTR). Additionally, the LSC region had the highest level of variation, followed by the SSC region, while that of IR regions was the lowest.

As inferred by IRscope, the gene number and order of all *Zingiber* plastomes were conserved; however, their LSC/IRs and IRs/SSC boundaries exhibited slight differences (Figure 3). The LSC/IRb boundaries were situated between *rpl22* and *rps19*, with a distance of 67–106 bp from the former gene. The IRb/SSC boundaries were 7–835 bp away from the *ndhF* gene in the SSC region. The *ycf1* gene was separated by the SSC/IRa boundary, with 93–5199 bp situated in the SSC and 309–5486 bp located in the IRa region. The IRa/LSC boundaries were 3–129 bp away from the *psbA* gene in 11 *Zingiber* plastomes, while they were overlapped in the plastomes of *Z. ellipticum* 4 and *Z. guangxiense*.

### 3.3. Microsatellites and Repeats

Using MISA-web, we have detected a total of 2855 microsatellites across all *Zingiber* plastomes, ranging from 78 (*Z. officinale*) to 113 (*Z. atroporphyreum*) in each species (Figure 5a). The mono-nucleotide type took the largest proportion of total microsatellites (44.94%), followed by di-nucleotide (28.65%) and tetra-nucleotide (18.64%) types, whereas the tri-, penta-, and hexa-nucleotide microsatellites accounted for 4.83%, 2.59%, and 0.35% of the total number, respectively. In addition, hexa-nucleotide microsatellites were only identified in plastomes of *Z. guangxiense*, *Z. mekongense*, *Z. purpureum*, *Z. roseum*, *Z. striolatum*, and *Z. wandingense*.

Totally, 1229 repeats were identified among 29 *Zingiber* species by REPuter analysis, from 30 (*Z. ellipticum*) to 79 (*Z. purpureum* 1) in each species (Figure 5b). Among all repeat types, the forward repeats (54.27%) appeared most frequently, followed by palindromic repeats (35.88%), while the complement and reverse repeats only accounted for 2.04% and 7.81% of the total, respectively. The complement repeat was not found in the plastomes of *Z. atroporphyreum, Z. longiglande*, *Z. mioga*, *Z. purpureum*, *Z. recurvatum*, and *Z. striolatum*. Additionally, *Z. purpureum* and *Z. recurvatum* did not contain reverse repeats.

### 3.4. Plastome Highly Divergent Regions

A total of 159 regions, including 89 genes and 70 IGS sequences, showed an aligned length over 100 bp and contained more than one mutation. According to the results of nucleotide diversity analysis, the *π* value of each gene or IGS region varied from 0.0003 to 0.0404 (Figure 6). Genetic divergence of IGS sequences was generally higher than that of gene sequences. Likewise, gene sequences located in the IR region exhibit significantly lower *π* values compared with those in the single-copy region (LSC and SSC). Six plastid regions with *π* greater than 0.02 were selected as highly divergent regions, namely *rpl20*, *clpP*, *ycf1*, *petA*-*psbJ*, *rbcL*-*accD*, and *rpl32*-*trnL*. Genetic distance analysis of the combined matrix showed a discrimination success of 86.21%, while that of each region ranged from 37.93% to 65.52% (Appendix A). In the most rapidly evolving regions (*clpP* and *rbcL*-*accD*), 91 and 194 variable base sites were detected, of which 47 and 174 informative base sites accounted for 2.11% and 13.71%, respectively.

### 3.5. Phylogenetic Relationships of Zingiber Species

Phylogenetic inferences based on a plastome dataset have obtained a reliable topology for *Zingiber* at the species level. As shown in Figure 7, the current *Zingiber* species formed a monophyletic group, which was sister to *Kaempferia*. Within the genus, *Z. ellipticum* from sect. *Pleuranthesis* is the most basal branch (BS = 100/PP = 1). The rest of the species were divided into two monophyletic clades (BS = 67, PP = 1). One contained species of sect. *Crytanthium* with highly supported values (BS = 100, PP = 1). The newly sequenced plastomes of *Z. cochleariforme*, *Z. guangxiense*, *Z. gulinense*, *Z. recurvatum*, *Z. wandingense*, *Z. mioga*, *Z. striolatum*, *Z. longiglande*, *Z. mekongense*, *Z. fragile*, *Z. simaoense*, *Z. yunnanense*, and *Z. roseum* belonged to this section. Another clade consisted of species from sect. *Zingiber* and sect. *Dymczewiczia* (BS = 100, PP = 1). *Zingiber atroporphyreum* was the only species we collected from sect. *Dymczewiczia*, and its phylogenetic position was nested in sect. *Zingiber*. The newly sequenced plastomes of *Z. officinale*, *Z. zerumbet*, and *Z. purpureum* belonged to sect. *Zingiber*. In addition, species with multiple accessions sampled can be grouped into monophyletic clades by plastome data, except *Z. mioga*, *Z. striolatum*, *Z. purpureum*, and *Z. montanum*.

## 4. Discussion

### 4.1. Plastome Structure and Characteristics Analysis

Plastomes have demonstrated great value in plant phylogeny primarily due to their relatively conserved structure and the exhibition of uniparental inheritance (maternal in angiosperms), providing unique insights into the contribution of seed dispersal to the genetic makeup of natural populations in comparison to nuclear markers [49,50]. In the present study, we newly assembled and annotated 29 plastomes for *Zingiber*. All of them displayed the typical quadripartite structure, similar genome size (161,495–163,880 bp), overall GC content (35.8–36.2%), and gene order. Consistent with previously published studies, we found *Zingiber* was relatively conserved, showing no gene rearrangement, replication, or loss [6,15,51], which suggested an enduring evolutionary stability within *Zingiber*. However, some variations were discovered at the SC/IR boundaries, which can be explained by the expansion and contraction of IRs, the main cause for length changes in angiosperm plastomes [52,53]. Changes in repeat sequences were another factor that influenced the genome size of the plastome. In this study, plastid repeat numbers ranged from 30 to 79 in each species, with forward (54.27%) and palindromic (35.88%) types appearing most frequently. Differences in repeat sequences among species were recognized as adaptations to environmental changes. A large number of repeats have a great influence on maintaining the structural stability of the plastome and promoting the generation of new genes [54,55,56]. A whole plastome identity plot again revealed an overall conserved feature in 29 *Zingiber* samples, with coding regions more conserved than non-coding regions. This could be caused by stronger natural selection forces in non-coding regions compared to those in coding regions [57]. In addition, nucleotide diversity analysis revealed a significantly reduced level of genetic divergence in IR regions compared to the single-copy regions (including LSC and SSC) as found in the previous plastome study of *Zingiber* [51], which may be influenced by copy correction occurring during gene transformation as well as the abundance of conserved rRNA genes in IR regions [58].

### 4.2. Plastome-Derived Markers of Zingiber

Plastid-derived markers have become valuable genetic resources, especially in the identification, protection, and breeding of some medicinal plants [32]. As an important economical plant genus in China, *Zingiber* faces a major obstacle to its identification, conservation, and utilization due to the lack of genetic resources. Due to the conservation of plastome structure and organization, plastid microsatellite primers are transferable across closely related taxa and are of great value in elucidating genetic diversity. Using MISA-web, a great number of plastid microsatellites were identified, with the proportion of mono-nucleotide type being the highest. The plastome-derived hotspot regions can provide a mass of genetic variations and were applied to produce a ‘high resolutive mini-barcode’ in plant identification [19,46]. Previous phylogenetic studies in *Zingiber* primarily relied on plastid *matK*, *rbcL*, *ndhC-trnV*, and *trnL-rpl32-ndhF* sequences, which often lacked sufficient phylogenetic resolution within closely related species [24]. Through nucleotide diversity analysis, we identified six hotspot regions (*rpl20*, *clpP*, *ycf1*, *petA*-*psbJ*, *rbcL*-*accD,* and *rpl32*-*trnL*) with *π* greater than 0.02. Although the number of mutations in these regions is far fewer than in the whole plastome sequence, they provide greater discrimination at the genus level than standard DNA barcodes. Among them, *ycf1*, *petA*-*psbJ,* and *rbcL*-*accD* have also been reported as candidate barcoding regions in previous studies of *Zingiber* [6,17,49], as well as in studies of other plants [11,59,60]. Given that the combined matrix of six hypervariable regions showed a higher discrimination success rate, we believe that they can be used as molecular markers to provide a substantial promise for the phylogenetic studies of *Zingiber*. In order to determine whether these gene sequences could serve as effective molecular markers, further exploration of the sequence size and the feasibility of primer design is imperative.

### 4.3. Intraspecific Phylogeny of Zingiber

Despite the important edible and medicinal values of *Zingiber* species, it remains a challenge to accurately distinguish close relatives or adulterants without any taxonomic capability due to complex inter-/intra-specific morphological variations within the genus. There are continuous safety-related problems reported globally caused by the inaccurate identification of herbal materials [61,62,63,64]. Thus, correct identification of plant materials is essential for the safety and efficacy of the foods and medicinal products.

Previous studies have made some efforts on *Zingiber* phylogeny using the methods of metabolic profiling [7], micromorphology [13], palynology [24], and DNA barcodes [2,24,65]. Among these methods, DNA barcoding technology has garnered considerable interest from researchers because it provides a convenient solution that does not require taxonomic knowledge or physiological conditions [66]. Standard DNA barcodes usually develop universal primers to extract short DNA sequences, which is beneficial for building databases and establishing a common identification criterion but shows unsatisfactory discrimination between the close taxa. Some standard DNA barcodes have been used in previous research to resolve the phylogenetic relationships of *Zingiber* but have not yielded satisfactory results [2,24].

On the basis of complete plastomes, we have provided a reliable phylogenetic topology for *Zingiber*. As shown by the phylogenetic tree, the current *Zingiber* species could be divided into three monophyletic clades, corresponding to sect. *Pleuranthesis*, sect. *Zingiber* + sect. *Dymczewiczia*, and sect. *Crytanthium*, as described based on inflorescence habit and pollen morphology [2]. Our results suggested that *Z. ellipticum* from sect. *Pleuranthesis* is the most basal branch, which is consistent with the viewpoint of Bai et al. [24] and Jiang et al. [6]. It is characterized by a peduncle arising from the side of the leafy stem, a lack of pulvinus, and spherical pollen with reticulate sculpturing [20,24]. The monophyletic sect. *Crytanthium* has radical inflorescences, procumbent peduncles, and ellipsoidal pollen with stripes on the surface [24]. In addition, in agreement with a previous phylogenetic study based on *ITS* sequences [2], our results indicated that sect. *Dymczewiczia* was nested within sect. *Zingiber*. Usually, the inflorescences of sect. *Dymczewiczia* occur apically on a leafy shoot, whereas sect. *Zingiber* develops radical, erect inflorescences [2]. However, previous studies have suggested that the two types of inflorescences (apical vs. radical) can occur within a species, such as *Z. officinale*, *Zingiber junceum,* and *Zingiber gramineum* [67,68]. The difference between these two types of inflorescences may be triggered by environmental factors; thus, they were not good diagnostic characters for classifying sections *Dymczewiczia* and *Zingiber* [67]. Moreover, both of them have spherical pollen with cerebroid sculpturing [23]. Therefore, we agree with Theerakulpisut et al. [2] to place sect. *Dymczewiczia* in sect. *Zingiber*.

Some species of *Zingiber* are not monophyletic groups in our study. For example, *Z. mioga* and *Z. striolatum* were entwined with each other with a low Bayesian posterior probability. The distribution of the two species overlaps in South China, and they have similar vegetative morphological characteristics but show different flower colors (yellow flower for *Z. mioga* vs. purple flower of *Z. striolatum*) [1]. Due to their wide distribution and considerable phenotypic variation, there may be several undescribed ecotype species. The division of complexes undoubtedly requires more genetic population research in future work. *Zingiber montanum*, *Z. purpureum*, and *Zingiber corallinum* are grouped into a clade. *Zingiber montanum* and *Z. purpureum* are frequently used to describe the Cassumunar ginger, a widely cultivated medicinal ginger [69,70]. However, Bai et al. [3] suggested that the correct name for Cassumunar ginger is *Z. purpureum*, whereas *Z. montanum* is a different species. Additionally, both *Z. montanum* and *Z. corallinum* are characterized by scarlet inflorescence. Therefore, these samples may be confused when collected. This also reminds us of the importance of accurate identification of sequencing samples for phylogenetic studies. Notably, *Zingiber* plants are widely distributed in tropical to warm-temperate Asia and contain extremely high species diversity. In order to understand their phylogenetic relationships and evolutionary history, future studies based on more comprehensive sampling strategies and rich molecular markers, such as nuclear genes, will be necessary.

## 5. Conclusions

The present study newly reports plastome information for 29 *Zingiber* samples. All plastomes displayed the standard quadripartite structure, similar genome size (161,495–163,880 bp), overall GC content (35.8–36.2%), and gene order. Through comparative plastome analysis, we found that IR expansions/contractions, as well as repeat variations, were the main reasons that influenced the genome size of the *Zingiber* plastome. A large number of SSRs and six highly variable regions have been identified, which can be used in future population genetics or phylogeography studies on this genus. A robust phylogeny of *Zingiber* with high bootstrap support was achieved using plastome sequences. It was strongly supported that current species of *Zingiber* were clustered into three clades, corresponding to sect. *Pleuranthesis*, sect. *Zingiber* + sect. *Dymczewiczia*, and sect. *Crytanthium*. Overall, this study solved the phylogenetic relationships of most *Zingiber* plants in China, and the newly reported plastome data and plastome-derived markers will be of great significance for the accurate identification, protection, and agricultural management of *Zingiber* resources in the future.

## Figures and Tables

**Figure 1 genes-15-01484-f001:**
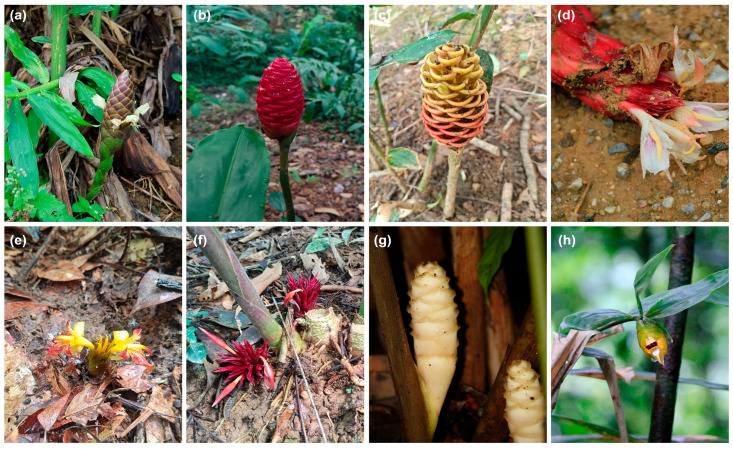
Inflorescence characteristics of some representative *Zingiber* species. (**a**) *Zingiber purpureum*; (**b**) *Z. zerumbet*; (**c**) *Zingiber spectabile*; (**d**) *Zingiber orbiculatum*; (**e**) *Zingiber teres*; (**f**) *Zingiber recurvatum*; (**g**) *Zingiber ellipticum*; (**h**) *Zingiber atroporphyreum*.

**Figure 2 genes-15-01484-f002:**
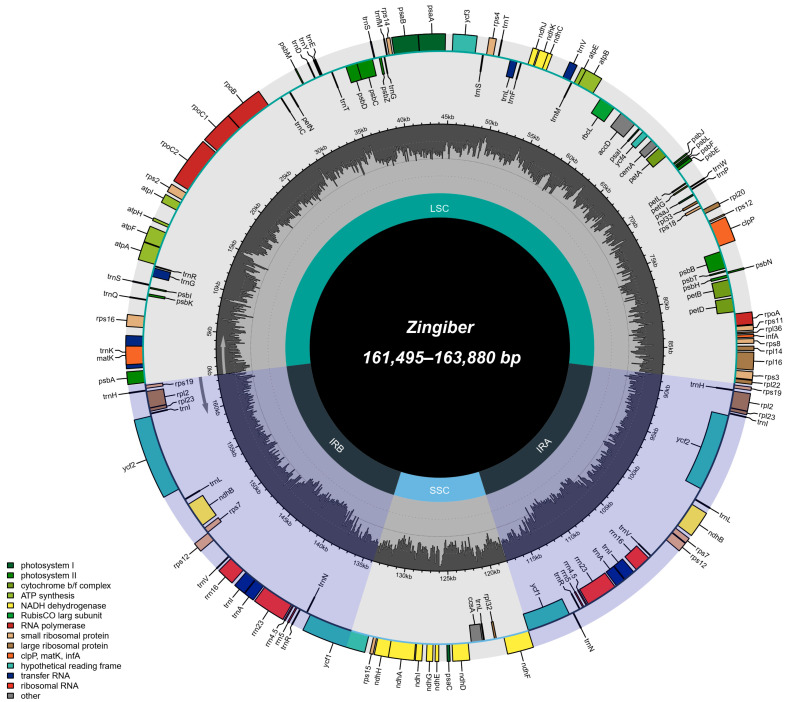
The plastome map of *Zingiber* species.

**Figure 3 genes-15-01484-f003:**
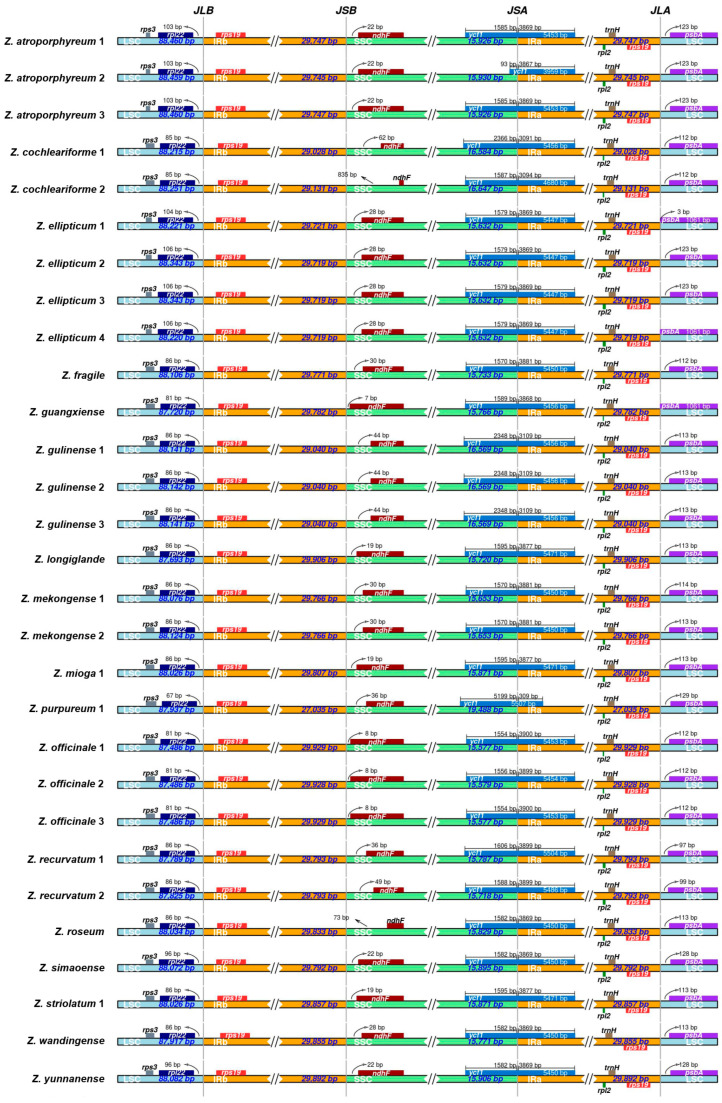
Differences of LSC, IR and SSC boundaries among *Zingiber* species.

**Figure 4 genes-15-01484-f004:**
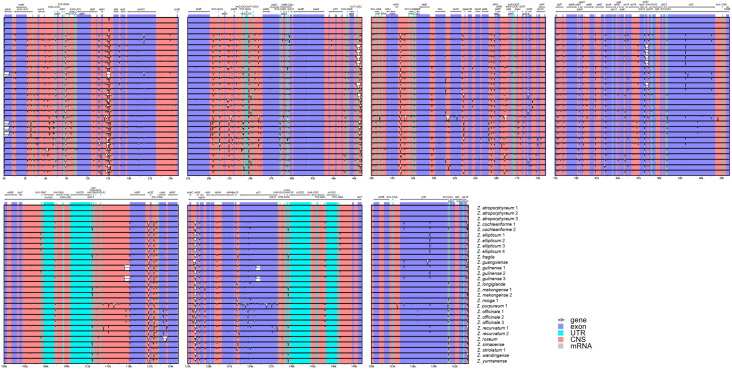
Sequence similarity plots among Zingiber plastomes. Annotated genes are shown along the top. The vertical scale indicates percent identity, ranging from 50% to 100%. Exons were colored by purple; untranslated (UTR) sequences were colored by blue; and conserved non-coding sequences (CNSs) were colored by pink.

**Figure 5 genes-15-01484-f005:**
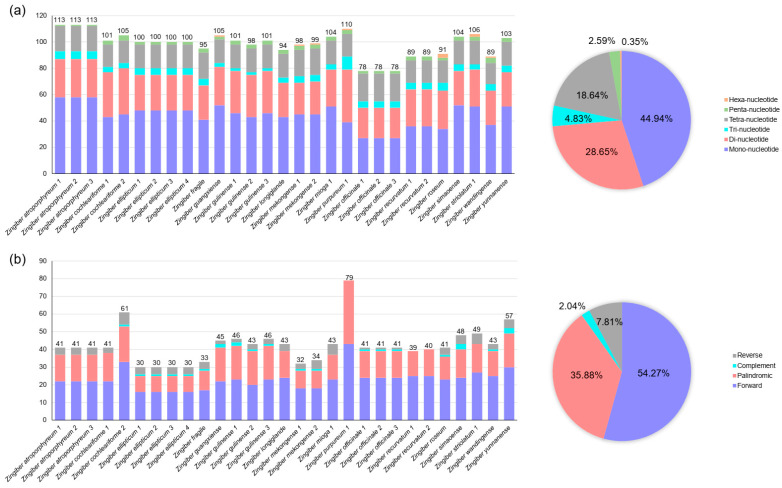
Characteristics of microsatellites and repeats among *Zingiber* species. (**a**) Numbers and proportions of microsatellites in different types; (**b**) numbers and proportions of repeats in different types.

**Figure 6 genes-15-01484-f006:**
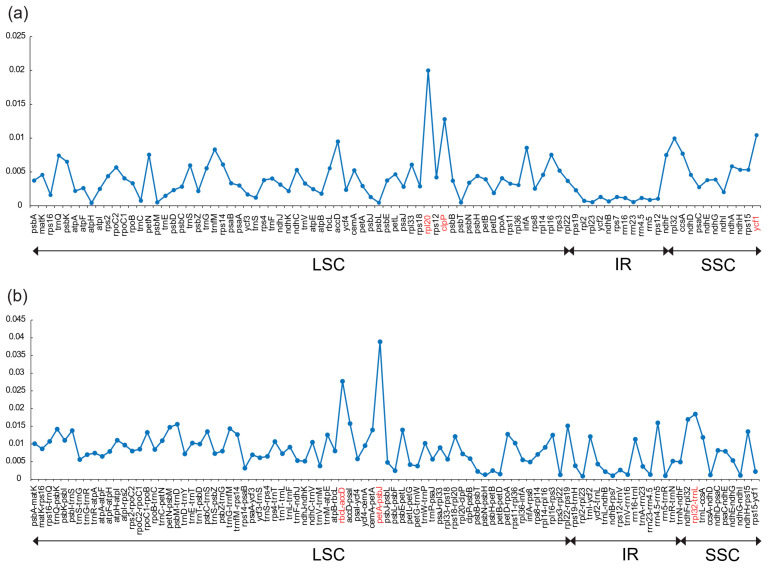
Nucleotide variability (*π*) of regions extracted from the alignment matrix of *Zingiber* plastome sequences. (**a**) *π* of 89 genes and (**b**) *π* of 70 intergenic spacers (IGS). Three genes (*rpl20*, *clpP*, *ycf1*) and three IGS regions (*rbcL*-*accD*, *petA*-*psbJ*, *rpl32*-*trnL*) exhibiting *π* values exceeding 0.02 were highlighted in red.

**Figure 7 genes-15-01484-f007:**
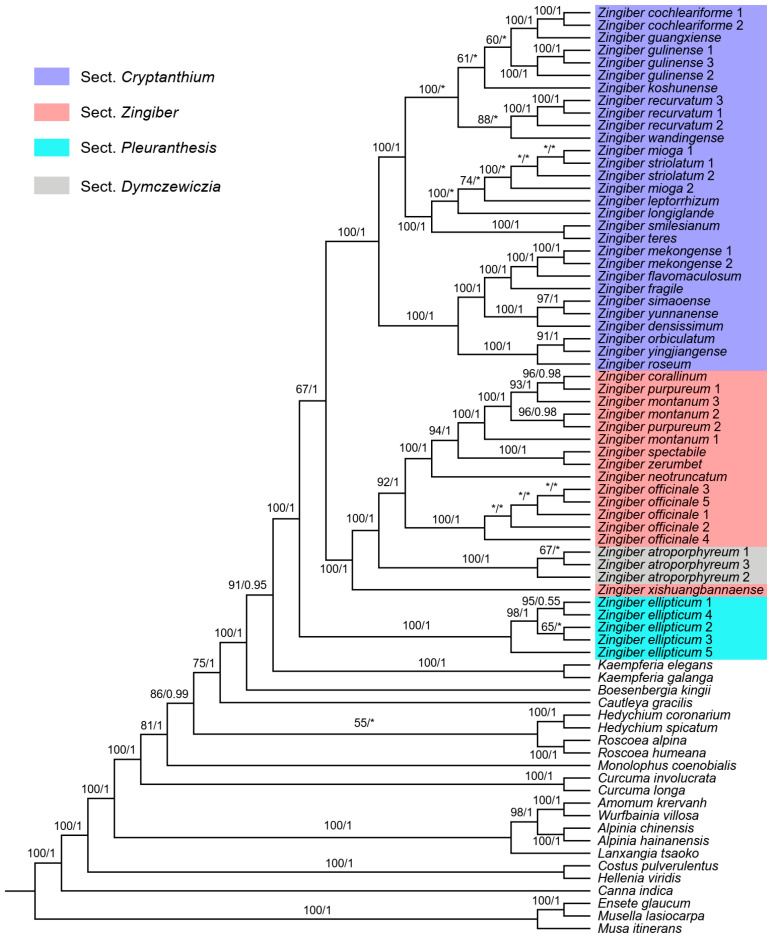
Phylogenetic trees of *Zingiber* based on complete plastome sequences. The tree shown depicts the ML topology with ML bootstrap support value/Bayesian posterior probability given at each node. Nodes with respective values less than 50/0.5 are marked as “*”.

**Table 1 genes-15-01484-t001:** Plastome information of the newly sequenced *Zingiber* species.

Taxon	Locations	Collection Number	GenBank Accession	Plastome Size (bp)	GC Content (%)
*Z. atroporphyreum* 1	China, Yunnan, Malipo	XMQ2023039	N_001486761	163,880	36.1
*Z. atroporphyreum* 2	China, Yunnan, Malipo	XMQ2023039-5	N_001486762	163,879	36.1
*Z. atroporphyreum* 3	China, Yunnan, Malipo	XMQ2023039-2	N_001486763	163,880	36.1
*Z. cochleariforme* 1	China, Guangxi	451223121026052LY	N_001486764	162,855	36.1
*Z. cochleariforme* 2	China, Guangxi	451223150119004LY	N_001486765	163,511	36.1
*Z. ellipticum* 1	China, Yunnan, Maguan	XMQ2023055-1	N_001486766	163,295	36.2
*Z. ellipticum* 2	China, Yunnan, Maguan	XMQ2023055-2	N_001486767	163,413	36.2
*Z. ellipticum* 3	China, Yunnan, Maguan	XMQ2023055-3	N_001486768	163,413	36.2
*Z. ellipticum* 4	China, Yunnan, Maguan	XMQ2023055-4	N_001486769	163,290	36.2
*Z* *ingiber fragile*	China, Yunnan, Puer	048956	OR337869	163,381	36.1
*Z* *ingiber guangxiense*	China, Guangxi	IBK00393893	OR337870	163,050	36.2
*Z. gulinense* 1	China, Yunnan, Maguan	XMQ2023054-1	N_001486770	162,790	36.1
*Z gulinense* 2	China, Yunnan, Maguan	XMQ2023054-4	N_001486771	162,791	36.1
*Z. gulinense* 3	China, Yunnan, Maguan	XMQ2023054-6	N_001486773	162,790	36.1
*Z* *ingiber longiglande*	China, Guangxi, Guilin	IBK00191773T	OR337871	163,225	36.1
*Zingiber mekongense* 1	China, Guangxi, Chongzuo	451402150915047LY-1	N_001486774	163,261	36.1
*Z. mekongense* 2	China, Guangxi, Chongzuo	451402150915047LY-2	N_001486775	163,309	36.1
*Z. mioga* 1	China, Hubei	00075720	OR337872	163,551	36
*Z. purpureum* 1	China, Yunnan, Malipo	XMQ2023048	N_001486776	161,495	35.8
*Z. officinale* 1	China, Yunnan, Qujing	S2	OR337873	162,921	36.1
*Z. officinale* 2	China, Hubei	S5	OR337874	162,921	36.1
*Z. officinale* 3	China, Chongqing	S17	OR337875	162,921	36.1
*Z. recurvatum* 1	China, Yunnan, Xishuangbanna	118587	N_001486777	163,162	36.1
*Z. recurvatum* 2	China, Yunnan, Xishuangbanna	42	OR337876	163,129	36.1
*Z* *ingiber roseum*	China, Yunnan	0425576	OR337877	163,529	36.1
*Z* *ingiber simaoense*	China, Yunnan	110728	OR337878	163,551	36.1
*Z. striolatum* 1	China, Chongqing, Jinfo Mountain	XMQ2023032-2	N_001486778	163.611	36
*Z* *ingiber wandingense*	China, Yunnan	49033	OR337879	163,398	36.1
*Z* *ingiber yunnanense*	China, Yunnan	0425625	OR337880	163,772	36.1

## Data Availability

The newly assembled plastomes have been deposited in GenBank (https://www.ncbi.nlm.nih.gov; accessions: OR337869–OR337880) and CNGB (https://db.cngb.org/; accessions: N_001486761–N_001486771, N_001486773–N_001486778).

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
