# Peer review of "Comparative Chloroplast Genome Study of Zingiber in China Sheds Light on Plastome Characterization and Phylogenetic Relationships"

_genes, 2024, doi:10.3390/genes15111484_

Round 1
Reviewer 1 Report
Comments and Suggestions for Authors
The research study clarifies phylogenetic relationships of genus Zingiber, using complete plastomes of 29 Zingiber accessions representing 17 Zingiber species from China. Samples are taken respectively Z. atroporphyreum, Z. cochleariforme, Z. ellipticum, Z. gulinense, Z. purpureum, Z. officinale, and Z. striolatum, from liquid nitrogen frozen fresh leaves. The other species are sampled from leaves of herbarium specimens. In the interpretation of the results and the phylogenetic analyses are included the published plastomes of Zingiberaceae and outgroups from GenBank . The paper is well written and illustrated. The results and analyses are convincing.
Comments
Do you find these paper suitable to compare with your results? https://doi.org/10.3390/plants8080283 and https://doi.org/10.3329/bjpt.v29i2.74398

Reviewer 2 Report
Comments and Suggestions for Authors
DearAuthors,
I have reviewed the manuscript and have the following observations:
The topic of the manuscript is plastomic analysis and phylogenetic analysis of intraspecific strains of Zingiber to develop and elucidate genomic resources. The topic of the manuscript is good and topical, Zingiber is indeed an important and increasingly used , multipurpose plant.
I think the manuscript is good, I only have problems with two chapters:
The Discussion chapter does not show the deeper context, so I suggest to rethink and rewrite it in this way and we cannot get an answer to how effective the measurement series was. I suggest moving the figures in the Discussion chapter to the Results chapter.
In the Concusions chapter, it would be worthwhile to mention the impact that the research and the results obtained have or could have in the future.

Reviewer 3 Report
Comments and Suggestions for Authors
The manuscript investigates the complete chloroplast genomes of 29 Zingiber accessions to elucidate their phylogenetic relationships and genomic characteristics. The study identifies significant variations in the plastome structure and provides a detailed phylogeny of the Zingiber genus. This research contributes to understanding Zingiber species' genetic diversity, aiding in their accurate identification and conservation.
-The introduction mentions Zingiber species' medicinal, edible, and horticultural values. Can you provide specific examples or case studies highlighting these species' economic and cultural significance in China and globally?
-The discussion on the commercial importance of Zingiber species should be expanded to include specific examples of their use in traditional medicine and modern applications.
-Can you provide more details on the criteria for selecting the 29 Zingiber accessions for plastome sequencing? Were these accessions chosen to represent the full diversity of the genus, or were there specific traits or regions you aimed to cover?
-Why did you choose the MGI-DNBSEQ platform for sequencing? Were there specific advantages or limitations of this platform that influenced your choice?
-You mention that previous studies have shown different phylogenetic placements for Zingiber sections. How does your study reconcile these discrepancies? What are the implications of your phylogenetic findings for the taxonomy and classification of the Zingiber genus?
-The study identifies six highly variable regions within the plastomes. Can you discuss the potential applications of these regions in more detail? How might these markers be used in future population genetics or phylogeographic studies?
-Based on your findings, what are the key areas of future research that should be pursued to understand the genetic diversity and evolutionary history of Zingiber species further? Are there specific technologies or methodologies you would recommend for these studies?
